# A rapid evaluation of quality of sedation and ventilation care processes for critically ill patients in Vietnam

An Luu Phuoc[1], Vo Tan Hoang[1], Truong Ngoc Trung[2], Nguyen Thien Binh[3], Vu Dinh Phu[4], Tran Minh Duc[1], Ha Thi Hai Duong[2], Tran Thi Diem Thuy[2], Duong Bich Thuy[2], Nguyen Thanh Nguyen[2], Le Thanh Chien[3], Doan Bui Xuan Thy[1], Nguyen Hoang Viet[5], Bui Ngoc Thanh[5], Vo Thi Hoang Dung Em[5], Jennifer Van Nuil[1,6], Abi Beane[7,8], Rashan Haniffa[7,8], Pham Ngoc Thach[4], Chau Minh Duc[5], Huynh Ngoc Hon[3], Nguyen Le Nhu Tung[2], Lam Minh Yen[1], Louise Thwaites[1,9], Duncan Wagstaff[10]*

**1** Oxford University Clinical Research Unit, Ho Chi Minh City, Vietnam, **2** Hospital for Tropical diseases, Ho Chi Minh City, Vietnam, **3** Trung Vuong Hospital, Ho Chi Minh City, Vietnam, **4** National Hospital for Tropical Diseases, **5** Dong Thap Hospital, Dong Thap, Vietnam, **6** Nuffield Department of Medicine, University of Oxford, United Kingdom, **7** Pandemic Sciences Hub and Institute for Regeneration and Repair, University of Edinburgh, Edinburgh, United Kingdom, **8** NICS-MORU, Colombo, Sri Lanka, **9** Centre for Tropical Medicine and Global Health, University of Oxford, Oxford, United Kingdom, **10** University College London, London, United Kingdom

* d.wagstaff@ucl.ac.uk

## Abstract

### Background

Sedation assessment, spontaneous awakening and breathing trials are evidence-based practices which can minimise harm from ventilation and sedation of critically ill patients. There are known difficulties in implementing these processes which are likely to be exacerbated in low-resource settings. This study aimed to describe current delivery of these care processes in three intensive care units in Vietnam; identify barriers and facilitators to their delivery; and describe local capacity for improvement.

### Methods

We conducted a prospective rapid evaluation between 01/11/2021 and 31/12/2023 comprising registry-enabled measurement of daily care processes, process mapping, observations, focus group discussions, semi-structured interviews and a structured assessment of local capacity for improvement. Contextual determinants of care quality were analysed using the Consolidated Framework for Implementation Research. Organisational capacity for improvement was analysed using the Model for Understanding Success in Quality.

**Data availability statement:** **APROS/ PROS at accept: Please follow up with AU to request revised statement which includes email address/URL where data request form can be submitted and requests for data placed** All data are available upon request to the OUCRU Data Access Committee (https://www.oucru. org/wp-content/uploads/2023/06/OUCRU-Data-Request-Form-V1.1-090217.pdf). This follows the standard policy of OUCRU, Oxford University and Welcome, to avoid exploitation of LMIC researchers and institutions and enable data compliance with both UK and Vietnamese law.

**Funding:** This research was funded by Wellcome [(224048/Z/21/Z), (107367/Z/15/Z), (089276/B/09/7), (217650/Z/19/Z)]. For the purpose of Open Access, the author has applied a CC BY public copyright licence to any Author Accepted Manuscript version arising from this submission. The funders had no role in study design, data collection and analysis, decision to publish, or preparation of the manuscript.

**Competing interests:** The authors have declared that no competing interests exist.

**Abbreviations:** RASS: Richmond Agitation-Sedation Scale; SAT: Spontaneous Awakening Trial; SBT: Spontaneous Breathing Trial; ICU: Intensive Care Unit; QI: Quality Improvement; CFIR: Consolidated Framework for Implementation Research; MUSIQ: Model for Understanding Success in Quality; STREAM: The Standards for Rapid Evaluation and Appraisal Methods; LMIC: low- and middle-income country; HIC: high-income country; CCAA: critical care in Asia and Africa; RAP: Rapid Assessment Process; GCS: Glasgow Coma Score.

## Results

Sedation was assessed qualitatively rather than using systematic tools. Spontaneous Awakening and Breathing Trials were both performed according to individual doctors' clinical judgement in a non-protocolised manner. Barriers to delivering these processes included the lack of locally-adapted protocols, perceived safety concerns exacerbated by staffing shortages and lack of familiarity due to confusing terminology. Facilitators to improvement included quality improvement champions, registry-enabled audit and feedback, training, and partnerships within and between hospitals.

## Conclusion

We identified opportunities to improve sedation and ventilation in the three study settings in Vietnam. The barriers to delivering the care processes we studied echoed those reported in high-income countries, but were exacerbated by local contextual factors such as staffing shortages and differences in professional roles. We developed recommendations for future improvement projects: implementing setting-adapted protocols, standardising terminology to improve documentation, engaging clinical staff with feedback, identifying champions, educate staff regarding the clinical processes and quality improvement and leverage existing internal expertise. These recommendations may have applicability to other care processes and/or settings.

## Introduction

Patients in intensive care units (ICUs) often require mechanical ventilation to maintain homeostasis, and sedation to minimise discomfort and maintain safety. However, unnecessarily prolonged periods of these interlinked processes can be harmful. Deep sedation is associated with longer durations of mechanical ventilation, which in turn are associated with ventilator acquired pneumonia, delirium, longer lengths of stay, and mortality. The Richmond Agitation and Sedation Scale (RASS) allows staff to systematically assess how sedated a patient is, by assigning a numerical score ranging from –5 to +5. It has shown to have high inter-rater reliability amongst diverse groups of medical professionals [1]. Maintaining a RASS score of –1 to +1 has been shown to reduce time to extubation and ICU length of stay [2]. Similarly, regular daily interruptions in sedation ('spontaneous awakening trials', SATs) can reduce time on ventilators, ventilator acquired pneumonia, venous thromboembolism and length of ICU stay [3]. Combining SATs with spontaneous breathing trials (SBTs) expedites liberation from mechanical ventilation, reduces time in hospital and all-cause mortality [4].

There are substantial barriers to implementing these care processes. Initial trials to demonstrate efficacy used relatively strict eligibility criteria and were resource-intensive to deliver [5]. Other barriers include: contra-indications in unstable patients,

or those with specific conditions; healthcare professionals without sufficient training; unclear or unusable protocols; and poorly coordinated multiprofessional communication [6].

These barriers may be exacerbated in low- and middle-income countries (LMICs). Financial and structural factors of ICUs vary according to national income levels, and hence strategies to improve care are also likely to vary [7]. Regular sedation assessment, SATs and SBTs have been reported to be implemented less frequently in LMICs than in high income countries (HICs) [8], and middle-income countries consistently report higher rates of ventilator acquired pneumonia [9]. Vietnam is a lower-middle income country, and its healthcare system, despite notable recent successes, faces many ongoing challenges, not least regarding underfunding. How sedation processes are currently implemented in Vietnam, and what determines this implementation, is not currently known. Furthermore, local organisational and contextual factors in individual ICUs are known to influence implementation of evidence-based care and capacity to improve service [10].

This study aims to: evaluate the quality of sedation processes in three Vietnamese ICUs; identify barriers and facilitators determining current care delivery; and assess organisational capacity for quality improvement (QI).

## Methods

### Study design

Understanding how complex processes such as sedation and ventilation are implemented in the ICU setting requires appropriate methodology [11]. We used a mixed methods rapid evaluation. Rapid evaluation methodology is used to identify priorities for and context to improvement of healthcare services in a timely manner [12]. The study design was informed by a rapid research decision tool [13]. The constituent methods we employed were: daily measurement of delivery of care processes, process mapping (to identify perceived gaps in quality), observations (to ascertain how care is actually delivered), focus group discussions (to consider the local context for improvement), semi-structured interviews (to understand individual perceptions of local systems). A structured assessment of local capacity for the Model for Understanding (MUSIQ) calculator [14] was also collected as the participating sites were planning to use the findings of this study to inform a future quality improvement project. A peer-reviewed protocol for the study has been published [15]. This manuscript has been written according to the Standards for Rapid Evaluation and Appraisal Methods (STREAM) [16].

### Setting

The study was conducted in three ICUs in Vietnam. Vietnam has population of more than 100 million people [17]. The country has made significant progress toward achieving Universal Health Care through enacting the Law of Social Health Insurance in 2008 and its revision in 2014 [18]. As of 2021, 89.3% of Vietnamese people had personal health insurance (via a national government scheme), up from 62% in 2010 [19]. This already high proportion is expected to increase beyond 95% by 2025 [20]. However, according to the World Health Organization, in 2016, out-of-pocket payments accounted for 41% of total health expenditure in the country [21]. The Ministry of Health manages three levels of health service delivery: primary level in districts and communes, secondary level in provinces, and tertiary level in national institutions under central government control. Private hospitals also exist with payment by private healthcare insurance schemes and/or out-of-pocket payments by patients [17].

The Collaboration for Research, Improvement and Training in critical care in Asia and Africa (CCAA) works with clinical and administrative stakeholders to establish capacity to evaluate and improve the quality of critical care. The collaboration has implemented a cloud-based intensive care registry enabling replicable capture of measures of quality and providing feedback through analytics dashboards and reports displaying indicators of quality [15]. The registry platform has evaluated as high quality according to frameworks of data quality, structural and organisational performance [22]. In Vietnam, seven ICUs joined the network, of which three participated in this study. ICU1 is a tertiary hospital and the other two ICUs

are provincial hospitals which manage patients with a variety of conditions. All three ICUs have an existing collaboration with CCAA and have established registry infrastructure to deliver audit and feedback.

## Ethics

Ethical approvals for the interviews were given by the Oxford Tropical Research Ethics Committee (OxTREC) in October 2020 (Reference: 558−20) and hospital IRBs in November 2021 (refs 3355/QD-BVBND). Data were collected between 01/11/2021 and 31/12/2023. Ethical approval was granted for the rapid evaluation in June 2021 (OxTREC reference 525−21) and IRB approvals in December 2022 (ref 4365/QD-BVBND). These data were collected between 01/02/2022 and 31/12/2023.

## Clinical processes being studied

RASS, SBT and SAT were chosen for study because local clinical leaders perceived sedation and ventilation to be an ongoing concern with staff and hoped to implement projects to improve them in the future. Unlike other quality processes, there were no Ministry of Health protocols describing RASS, SAT and SBT, and therefore different approaches were taken at different hospitals. The processes also had a strong evidence base, and had been recommended as feasible, reliable, valid and sensitive by a consensus-building study investigating critical care quality indicators in LMICs [23]. Definitions for the care processes can be found in Supplementary Information.

## Eligibility, sampling and recruitment

Staff were considered eligible to participate if they were aged 18 years or older, were involved in the design and/or delivery of care for ventilated and/or sedated patients in the ICU and if they provided informed consent (in writing for interviews and FGDs, or verbally for observations) after receiving a participant information sheet (Supplementary Information). Patients were included in the registry data if they were admitted to ICU for at least 24 hours, received mechanical ventilation and/or sedation, and were aged 18 years or older. Participants were purposively sampled [24] to reflect a range of professional, clinical and organisational views. participate as they were not involved in the sedation-related processes of care being studied. Sampling continued until theoretical saturation was achieved [25].

## Data collection

We conducted two cycles of data collection including process mapping, structured observations, focus group discussions and MUSIQ questionnaire completion. All data collection with participants was conducted in Vietnamese before subsequent translation into English. Procedures for collecting and reporting daily quantitative data via the registry to describe delivery of clinical processes are already published [22].

Process mapping was conducted during in-person sessions with between two and four multidisciplinary clinical members staff [12] (Supplementary Information). Participants created visual maps of patient pathways for the study processes (RASS, SAT and SBT) to represent the current typical process, preferences for how it should ideally be conducted. Contextual determinants influencing the study processes were integrated into the maps.

Structured observations were conducted in settings and scenarios where sedation and ventilation were being planned, delivered or discussed. Settings were identified prior to the start of data collection by consultation with ICU teams and adjusted based on findings during the study. Data were collected and contemporaneously documented by ALP using a structured observation template to ensure data consistency (Supplementary Information).

Focus group discussions were conducted with between two and eight participants. Discussions were semi-structured, using a guide (Supplementary Information) to maintain consistency between settings, but responsive to emerging themes and findings from the process mapping and observations. Audio recordings of discussions were collected and contemporaneous field notes made.

The MUSIQ questionnaire is a validated tool to identify organisational experience of and capacity for quality improvement [14]. The questionnaire was completed at each ICU at the beginning and end of this project. Data collection was coordinated by a member of the evaluation team with 3–6 participants agreeing consensus answers about local organisational contextual factors. The total possible score is 168, and there are seven thresholds which suggest the likelihood of success of QI projects given the context (Supplementary Information).

Semi-structured interviews were carried out to gain deeper insight into the processes of audit and feedback within the ICUs. These followed pre-designed interview guides and were conducted by research staff trained in social science methodology. Participants were interviewed twice: at the beginning of the study and again approximately 12 months later.

## Data analysis

Data collection and interpretative analysis occurred in parallel. Analysis was coded according to the Consolidated Framework for Implementation Research (CFIR) framework [26]. The findings of the analysis were iteratively discussed by the study team, and then reconstructed using the Rapid Assessment Process (RAP) sheet (Supplementary Information), to maintain consistency and discover overarching themes. The validity of findings was discussed and checked with local clinical and registry participants at every site. Triangulation was performed by comparing and contrasting findings between qualitative methods, and with the registry data. Patients or the public were not actively involved in the design or conduct of this study.

## Evaluation team

The team comprised four members to provide sufficient breadth of research expertise, language skills, and clinical experience in critical care. Three members were based in Vietnam, two of whom in a research capacity, and one in a combined clinical/research role at one of the study hospitals. One native Vietnamese speaker collected all qualitative data face to face in the study settings. One team member had undergone training in rapid evaluation methods, and three had extensive experience of qualitative methods. A reflexivity statement informed data analysis (Supplementary Information).

## Results

Three ICUs participated in the study (Table 1), of which one was a tertiary hospital in an urban setting, and two were provincial (one urban, one rural). Registry data showed that tetanus was the most common diagnosis in ICU1, and that sepsis and pneumonia common across all three ICUs. ICU3 used mechanical ventilation for a lower proportion of patients than the other ICUs.

A total of 34 participants were included in the study, including 27 doctors and seven nurses (Table 2). Structured observations were conducted during ward rounds, handover sessions and morning nursing care sessions as this was when sedation was discussed and acted upon. The interviewees were the doctors and nurses that collected data for the registry.

**Table 1. Participating hospitals & ICUs.**

| Hospital name & location | Participating ICU | ICU beds | Hospital beds | Hospital description | Top 3 diagnosis | % patients on MV |
|---|---|---|---|---|---|---|
| ICU1 | Medical ICU | 30 | 550 | Tertiary hospital, Urban setting | Tetanus Pneumonia Sepsis | 65.6 |
| ICU2 | Medical ICU | 40 | 700 | Provincial hospital, Urban setting | Respiratory diseases, CVD, sepsis | 67.3 |
| ICU3 | Medical ICU | 40 | 700 | Provincial hospital, Rural setting | Pneumonia Sepsis ARDS | 46.8 |

**Table 2. Individual participants in qualitative data collection.**

| ICU | Process Mapping | Observations | MUSIQ | Focus groups | Semi-structured interviews |
|---|---|---|---|---|---|
| ICU1 | 2 doctors, 2 nurses | Morning ward round x5, Shift handover x2 Morning nursing care x2 | 2 doctors, 1 nurse | 1 doctor, 1 nurse | No interview, change in staff |
| ICU2 | 2 doctors, 1 nurse | Morning round x2 Morning nursing care | 2 doctors | 2 doctors, 1 nurse | 3 doctors |
| ICU3 | 2 doctors | Morning rounds Morning nursing care | 2 doctors | 7 doctors, 1 nurse | 2 doctors |

## Description of current processes and perceived areas for improvement

**RASS.** As summarized in Table 3, health professionals used qualitative labels for assessing sedation (such as 'irritated/ awake/ asleep/ sedated') much more commonly than quantitative or categorical scores such as RASS. Sedation, or more commonly 'consciousness', was discussed during ward round discussions in ICU1 and ICU2, but these rounds and discussions were less frequently observed in ICU3.

There were parallel systems of assessment of sedation between doctors and nurses, using different metrics, and for different purposes. Nurses typically assessed and documented consciousness (using the Glasgow Coma Score (GCS) every 3 or 6 hours in all the ICUs. A project to improve documentation of RASS was being performed at ICU1, during which training was being delivered to nursing staff to increase familiarity with the tool. This led to RASS being documented on a dedicated audit sheet, separate to the nursing record, where it was not routinely viewed, discussed or used during normal daily care. These parallel systems explain the disagreement between observation findings (e.g., RASS wasn't being used) and registry data indicating compliance of 83% patient days. In ICU2, doctors documented their assessment of RASS in their clinical notes, which were rarely discussed or accessed by nurses.

In all the ICUs, only doctors adjusted sedation dosing to achieve the depth of sedation they were seeking. Junior doctors were required to have their plans and orders approved by senior doctors. Nurses were not allowed to adjust sedation dosing (or any other medication) because of regulations to what they can practice in Vietnam; instead, they would report problems with sedation to doctors.

**Table 3. Quality of current care processes in ICU.**

| Indicator | ICU | Process compliance | Current process | Opportunities for improvement |
|---|---|---|---|---|
| RASS | ICU1 | Target 83% Actual 83% | Sedation (mostly) described qualitatively Targets not set Patients deeply sedated | Minimise duplication of documentation Improve communication between doctors and nurses |
| | ICU2 | Target 17.6% Actual 17.6% | | |
| | ICU3 | Target 21% Actual 21% | | |
| SAT | ICU1 | 0% | Sedation only titrated by doctors 'Weaning' guided by individual doctors' judgement at ICU 1 & 3 Local guideline used at ICU 2 | Clarification of terminology Reduce variation between clinicians Improve consistency of documentation |
| | ICU2 | 33.6% | | |
| | ICU3 | 17.7% | | |
| SBT | ICU1 | 9.6% | 'Weaning' guided by individual doctors' judgement Dedicated data collection to facilitate audit and handover at ICU2 | Clarification of terminology Reduce variation between clinicians Standardise communication (e.g., at handover) |
| | ICU2 | 48.4% | | |
| | ICU3 | 32.5% | | |

Many patients were deeply sedated due to the use of paralysing medications and/or their underlying disease (e.g., tetanus).

Documentation of sedation dose changes were made by doctors in the drug charts and/or medical records. Nurses hoped that these decisions could be communicated to them more consistently, and that using RASS might enable this. Doctors described strong personal ownership of sedation management for their patients, and this ownership was transferred during shift handovers to the next shift doctor.

*'I am in charge of that person... I am the one who set the [RASS] target. And if the shift changes on that day, a new doctor will examine that patient and set his/her own target for that day at the beginning of the shift'. [ICU3 FGD1]*

**SAT.** The term 'SAT' was not used at any of the ICUs. The concept of 'sedation breaks' was more familiar and used occasionally in ICU1 (for selected patients who had only received short durations of sedation) and more commonly in ICU2 (48.4% eligible patient days). This was because ICU2 was the only setting which had a specific protocol for conducting 'sedation breaks'. This included a safety checklist to determine patients' eligibility for a sedation break, and a protocol for how they should be conducted. Doctors in this ICU documented plans and progress in an online spreadsheet accessible to other doctors, which facilitated effective sharing and handover of information to their colleagues. The protocol required the application of individual clinical judgement therefore its use varied between doctors:

"[The decision to perform SAT] *depends on the evaluation of each doctor. It's true that we follow the guidelines but the clinical evaluation of doctors [are] based on their experiences so SAT can be a little early or delayed.' ICU 2]*

Participants commonly described sedation 'weaning' at all three ICUs, describing the gradual reduction of sedation by a small amount each day. Not all the sedation weaning attempts were documented, especially if they were unsuccessful, making it harder for other staff on other shifts to track patients' progress. A large number of patients were not suitable for weaning because of contra-indications due to their underlying diseases.

Sedation was typically not weaned on weekends or holidays (or overnight) due to fear of accidental extubation (patients pulling their breathing tubes out – a medical emergency) when there were fewer senior staff present, as exemplified by this FGD participant:

*"[We're] trying not to wean in the evening or on holidays, I'm afraid we can't monitor closely enough when we don't have office hour staff, only duty staff, it's easy to make mistakes when we don't monitor close enough." [ICU 3]*

**SBT.** Participants did not refer to the term SBT, but rather called it 'weaning from ventilation'. This process involved gradually reducing the support that the ventilator provided to the patient's breathing, including switching the mode of the ventilator from mandatory to spontaneous. There were no standardised criteria at any of the ICUs for choosing which patients to trial this in, or when, or defining whether it was successful. Instead, decisions were made according to individual clinical judgement of the doctors on duty. Nurses were not allowed to adjust ventilator settings according to national regulations in Vietnam. Doctors tended to only document progress if it was successful, and it was usually only performed during weekdays. ICU2 had a shared document where doctors updated patient information (including weaning details) to hand over to doctors of subsequent shifts. This standardised form of handover did not happen at ICU1 or ICU3.

## Determinants of care quality

Contextual determinants of care quality are summarised according to the Consolidated Framework of Implementation Research (CFIR) [26] in Table 4, and are reported below.

**Table 4. Determinants of care quality according to the Consolidated Framework of Implementation Research (CFIR) [26].**

| | Barriers to care quality | Facilitators of high care quality |
|---|---|---|
| Intervention | Relative advantage<br>Costs | Adaptability |
| Implementation processes | Teaming | Engaging<br>Reflecting & Evaluating |
| Outer setting | Policies & Laws | Partnerships & Connections<br>Financing<br>External pressures |
| Inner setting | Communications<br>Compatibility | Relative priority |
| Individuals | Capability<br>Motivation | Opinion Leaders |

### Intervention (key attributes of the clinical processes under study: RASS, SAT ad SBT)

Clinical staff reported not being persuaded of the *relative advantages* of using RASS, SAT and/or SBT. For example:

> *'We are not proactive in [measuring RASS] it because we don't see the benefits.'* [nurse, ICU1, FGD]

Mostly this was due to a lack of familiarity with these terms and/or processes, leading to persistence of existing care processes (such as qualitative descriptions of consciousness). The exception to this norm was in ICU2, where the clinical lead had instituted local protocols for measuring RASS and conducting SATs and SBTs. The other ICUs did not have protocols for sedation assessment.

There were only two examples of active opposition to using these processes. Firstly, there was reluctance to measure RASS as it was (erroneously) perceived to increase workload (*'costs'*) by duplicating existing processes, such as assessment of consciousness. Secondly, at ICU1, doctors felt SAT was not helpful for specific groups of patients, for example those with tetanus. Hence participants felt that the processes would need *adaptations* to suit this different approach to weaning of sedation.

### Implementation process (i.e., how RASS, SAT and SBT were implemented locally)

An in-person workshop had recently been held by CCAA to help *engage* staff at all three ICUs in improving sedation processes. The evidence base for the clinical processes being studied had been explained and participants had received training in quality improvement techniques, including process mapping.

Audit and feedback, via the registry platform, was planned to evaluate local delivery of RASS, SAT and SBT. Data collection processes varied between the ICUs. At ICU1, non-clinical staff collected data. These staff were not directly involved in routine clinical care of patients, and struggled to interpret clinical documentation; for example, documentation of 'sedation weaning' made it hard for them to ascertain whether SAT and SBT had been delivered. In contrast, clinical staff at ICUs 2 & 3 collected registry data, forcing them to *engage* with the data as they collected it. Furthermore, data completeness at all ICUs was regularly reviewed by a data quality officer, who highlighted gaps to all local sites. It is therefore possible that some data were entered retrospectively.

Challenging data collection inhibited the opportunity for *reflection and evaluation* which monthly feedback (via registry reports) provided. Data were discussed at monthly meetings, but attention was focused on technical issues of data completeness, or metrics which had to be reported externally (such as those describing operational issues such as admissions

and discharges, or hospital acquired infections which the department of health needed reporting weekly) rather than conduct of the interventions themselves. Despite these challenges, interviewees at ICUs 2 & 3 stated that the reports helped them to look at sedation processes, and was helping them to make changes: *"We prioritize to stop sedation early to prevent VAP since we have Registry."*

**Outer setting (the context(s) in which the ICUs exist, e.g., the hospital, community and/or country)**

*Partnerships* within the Vietnam network of ICUs, and with the registry team, facilitated implementation through training, registry infrastructure, resources for quality control, IT support and funding for data collectors. The three hospitals prioritised using the registry to obtain data aligned with *external pressures* for *performance measurement* by the Department of Health. There was a lack of overt *financing* incentives (or penalties) regarding sedation and ventilation from insurance companies (the most common healthcare payers in Vietnam) in contrast to countries like the USA. Since clinical records were designed to provide data that insurance companies required for payment, fields for RASS, SAT and SBT were omitted. Consequently, when ICU1 started recording RASS, a parallel set of medical records was created, creating duplication of work. *National policies* also influenced implementation because nurses were not authorised to manage doses of sedation and/or ventilator settings (or other medications).

**Inner setting (the ICU settings)**

*Communication* of sedation processes worked well within professional silos (i.e., between doctors during ward rounds, and between nurses at their handover meetings). However, nurses felt doctors didn't always communicate sedation and/or ventilation weaning plans to them. For example, junior nurses were sometimes left confused:

> *'Some doctors don't tell the nurses that they are doing SBT; experienced nurses can notice [changes] from the ventilation mode, but junior nurses observe differences on patients and will ask the doctors.' [ICU1]*

At all three ICUs, there were 7–8 nurses looking after between 30–40 patients, of whom 45–65% were mechanically ventilated (Table 1). These high patient:nurse ratios increased fears of safety incidents if patients were not adequately sedated. These safety fears were exacerbated at night when fewer doctors were present. Furthermore, at ICU1, the night shift of doctors came from departments outside ICU, and so were less familiar with sedation processes. Participants felt that the variable level of experience of these doctors reduced the *compatibility* of using RASS, SAT and SBT consistently. Standardising sedation processes was more advanced at ICU2 then the other two settings, and as such it had greater *relative priority* there.

**Individuals (the roles and characteristics of individuals involved)**

*Opinion leaders* had a large influence on how sedation was performed, but the individuals who fulfilled these roles, and their impacts, varied between sites. At ICU2, the clinical lead of the ICU was proactive in implementing QI methods. ICU2 doctors recorded sedation and ventilation parameters in a dedicated spreadsheet to help them decide whether to initiate sedation breaks and/or wean ventilation; these data were also used for audit and research projects. He had also instigated local guidelines for sedation and ventilation weaning (which other ICUs were keen to borrow and implement in their own settings). He also ensured that sedation plans were discussed and documented during doctors' handover meetings. Similarly, at ICU1, the clinical lead was perceived as influential, and was keen to improve documentation of RASS in parallel with this evaluation and delivered training sessions to nurses and doctors. We didn't identify an opinion leader at ICU3.

There was variation between staff, but most participants agreed there was *motivation* to improve sedation and weaning processes. Staff perceptions of their own, and their peers', *capability* was more mixed. Training was delivered at ICU2

for doctors and nurses to follow the local sedation guideline consistently. Only staff at ICU2 were familiar with using QI methods.

## Organisational capacity for improvement

The results of the MUSIQ questionnaires are shown in Table 5. Scores at ICUs 1 & 2 indicate that a QI project focused on the three care processes under study would have a reasonable chance of success, and possible contextual barriers existed at ICU3. ICU2 had the highest score of 150 which tallies with the qualitative findings above regarding the influence of the QI Champion, presence of protocols and clinician familiarity with QI methods at that unit. The registry-enabled audit and feedback was important infrastructure that staff saw supporting implementation. Participants perceived that further engagement from departmental and hospital leaders would be necessary to support implementation. All three hospitals had dedicated general planning and quality management departments, mandated to work on priorities set by national policies. They possessed the skills (and some data) to measure and improve sedation and ventilation, but were disconnected from the clinical teams attempting to improve these care processes. This represented a missed opportunity to leverage r*elational connections* within the hospital. Scores at all three ICUs remained within the same brackets at the end of the study in comparison to the start, suggesting there weren't material changes in context during the period of data collection.

## Discussion

This is the first evaluation of current delivery, and barriers and facilitators, of three evidence-based care processes for sedated and/or ventilated critical care patients in Vietnam. Sedation was not being routinely assessed using RASS or any other systematic tool. Instead, staff were evaluating patients' sedation qualitatively, or more commonly their level of consciousness. SATs (called 'sedation breaks') and SBTs (conceptualised as part of ventilation weaning) were both performed according to individual doctors' clinical judgement in a non-protocolised manner. Barriers to delivering these care processes included the lack of locally-adapted protocols, perceived safety concerns exacerbated by staffing shortages and lack of familiarity due to confused terminology. Facilitators included QI champions, registry-enabled audit and feedback, training in both clinical processes and QI methods, and partnerships within and between hospitals.

Clinical staff were not convinced of the benefits of using RASS to set sedation targets. Partly this was lack of familiarity with the process, but partly because the intended benefits (delegating autonomy for titrating sedation to the bedside nurses) did not apply to the Vietnamese setting due to restrictions on nurses adjusting medication. The evidence base for the benefits of using RASS largely comes from HICs where nurses have the authority to titrate sedation. It's likely therefore that any protocol borrowed from overseas would need to be substantially adapted to suit the Vietnamese system, workforce, and patient population. These issues apply to SAT and SBT as well: research in HICs has shown that frontline staff agree that the routine use of these processes expedites extubation in comparison to methods relying upon individual clinical judgement [27], but participants in our study were less convinced. This may explain part of the difference in compliance that we found: in the USA (in a trial setting) SBTs were performed on 85% eligible patient days in comparison to 10–48% that we observed. Tools, such as ADAPTE (adaptation framework & resource toolkit) exist to aid translation of clinical practice guidelines between different settings and populations [28].

**Table 5. Organisational readiness for quality improvement (MUSIQ score).**

|  | Start of study | End of study | Change | Mean | Interpretation |
|---|---|---|---|---|---|
| ICU1 | 142 | 125 | −17 | 133.5 | QI project on RASS/SAT/SBT has a reasonable chance of success |
| ICU2 | 150 | 150 | 0 | 150 | QI project on RASS/SAT/SBT has a reasonable chance of success |
| ICU3 | 119 | 88 | −31 | 103.5 | QI project on RASS/SAT/SBT could be successful, but possible contextual barriers |

Staff perceived that SATs and SBTs weren't always compatible with existing workflows. For example, there was tacit reluctance to perform SATs and SBTs due to perceived safety concerns, primarily risks of patients removing their own breathing tubes. This reluctance mirrors fears from research in high-income settings [29], where there is wide variation in practice [30–33]. The literature indicates that common clinician concerns regarding SATs include potential patient discomfort, distress and safety [30]. Conducting SATs or SBTs requires staff to remain with patients for extended and/or unpredictable periods of time limiting their ability to complete other tasks [27]. The literature describes how many competing priorities can therefore take precedence over or interrupt SAT/SBT plans [27]. Nurses desired better communication of sedation plans from doctors. Interprofessional coordination of sedation and ventilation weaning is inherently difficult and requires anticipating other team members' behaviors [27,34]. Co-designing ICU processes with local stakeholders has been done successfully by our group in Vietnam [35], and may help adapt SAT/SBT to suit existing workflows.

These barriers and risks can be offset through closer monitoring of patients to maintain patient safety. This has been found to be possible in high-income settings [4] and in trial settings where additional research staff are typically present [29]. The additional workload for healthcare professionals, who are already in short supply in Vietnam, remains a barrier. The nurse:patient ratio in the ICUs we studied was approximately 1:4–6 (of which approximately 50% patients were mechanically ventilated), which is much higher than the 1:1–2 recommended in HICs. Previous research has shown that a lack of nurses is an obstacle to ICU function in LMICs [7]. Increasing nursing workforce is unlikely to be achieved quickly, but improving training and skills (another commonly cited barrier to QI in LMIC ICUs [7] can be achieved [36].

We found a lack of clarity of terminology describing sedation and ventilation. Sedation assessments were confused with assessments of consciousness; SATs were also named sedation breaks, and also confused with longer term weaning of sedation; and SBTs were confused with ventilation weaning. Previous studies have found that clinical staff conflate SBTs with spontaneous breathing modes and weaning ventilatory support generally [27]. Clarifying how these terms were conceptualised and used by participants in our study was even harder when they needed translation between English and Vietnamese. Sometimes the multiple terms might reflect confused understanding, but sometimes they reflected deliberate interpretations; for example, a previous study found that some nurses preferred to gradually reduce the rate of sedative infusion rather than pause it completely, because of perceived safety concerns [27]. This multiplicity in terminology was flourishing in the absence of national standards or definitions, leading to variation in interpretations at individual and ICU levels. Clarifying terminology would help standardise communication, reduce the multiple parallel threads of clinical documentation, increase staff understanding of other professionals' roles and facilitate building shared mental models for using RASS/SAT/SBT [27]. Clarification might be assisted by the development of local protocols, which have been shown to promote implementation of sedation processes [8], including in LMIC ICUs [7,8,37]. Participants in our study enjoyed using the protocol which had been developed at ICU2, and other ICUs were keen to adopt this protocol. This finding is likely to have applicability beyond Vietnam: a worldwide study in 2021 reported that written protocols were only present in 49%, 35% and 47% of ICUs for RASS, SAT and SBT respectively [38].

There is an opportunity to use the registry platform for audit and feedback. ICU registries are validated and effective methods of quality improvement in LMICs [7], and have previously been suggested as a strategy for assisting implementation of sedation protocols [27]. This approach would facilitate evaluation and improvement of quality but requires agreed metrics (and definitions) and is only useful if feedback is disseminated and discussed to prompt reflection. This can be easier said than done. We found problems with data collection, and a lack of dissemination of feedback. Previous studies have found that already busy frontline clinical leaders have struggled to find capacity to identify opportunities for improvement [5]. This may well be another opportunity for productive collaboration between hospitals [7].

Local champions and opinion leaders were influential in local delivery of care. There was a markedly differential approach to sedation and ventilation at ICU2 where a champion had galvanised data collection and instituted protocols. Identifying and supporting champions is recognised to be an effective way of driving change. But it is unfeasible to rely upon these individuals in the longer term, as ICUs are typically not staffed with sufficient personnel dedicated to facilitating

adherence or leaders trained in implementation of evidence-based care [5,39]. Instead, QI needs to be institutionalised within clinical departments. How this is done will depend on the context of the ICU; previous research has shown that sedation practices depend upon the success (or failure) of previous interventions, clinicians' age, training, size of ICU, and whether they practice in a university-affiliated hospital [27,30]. Furthermore, national income level, use of a multidisciplinary approach and presence of specialist intensivist was associated with implementation of sedation protocols [8]. In our study, using the opportunity to work with hospital quality departments seems a sensible way to increase organisational capacity for improvement.

---

**Box 1: Recommendations for future quality improvement projects aiming to improve sedation and ventilation**

1. Agree setting-adapted protocols for the care processes to be improved

2. Use the protocols to clarify terminology regarding the processes, thereby facilitating better documentation and auditing of quality

3. Ensure audit data is fed back to and discussed by frontline clinical staff

4. Use training and education to persuade clinical staff of the relative benefits and safety of the care processes

5. Identify and support Champions to drive improvement

6. Provide training for clinical staff on quality improvement

7. Build relationships with existing quality departments to leverage their capacity and expertise

---

### Limitations

Firstly, this study was limited to ICUs in government hospitals in Vietnam and should not be considered representative of practices in other settings. However, by describing the contexts of the ICUs we studied, we hope readers may be able to judge how our findings apply to their own settings. Secondly, by triangulating interview and focus group findings with observations and process data we have minimised but not completely removed the effects of our institutional alignments. Thirdly, we considered three specific clinical processes, but of course these are not performed in isolation but are inextricably linked to multiple other clinical processes and patient outcomes which were beyond the scope of this study. Fourthly, the sample of clinical staff that contributed data to this study necessarily limits the generalizability of our findings. Finally, a hangover from the COVID-19 pandemic may have been present during this study (conducted in 2022−23). The pandemic influenced patient acuity, workload (such as prone positioning) and workforce constraints (such as use of personal protective equipment) and may have rendered fewer patients eligible for SAT/SBT [27].

### Conclusion

We identified opportunities to improve the ventilation and sedation of critically ill patients in Vietnam. The barriers to consistently using RASS, SAT and SBT echo those reported in high-income countries, but were exacerbated by local contextual factors such as staffing shortages and differences in professional roles. We developed recommendations for future improvement projects targeting RASS, SAT and SBT in Vietnam: implementing setting-adapted protocols, standardising

terminology to improve documentation, engaging clinical staff with feedback, identifying champions, persuade staff of the relative benefits, provide QI training, and leverage existing internal expertise. These recommendations may have applicability to other care processes and/or settings.

## Supporting information

**S1 File. Supporting information files contains S1-S9 Files.**
(DOCX)

**S2 File. Inclusivity-in-global-research-questionnaire.**
(DOCX)

## Acknowledgments

Data collectors. Fathima Fazla.

## Author contributions

**Conceptualization:** An Luu Phuoc, Thuy Thi Diem Tran, Abi Beane, Rashan Haniffa, Lam Minh Yen, Louise Thwaites.

**Data curation:** An Luu Phuoc, Tran Minh Duc, Duncan Wagstaff.

**Formal analysis:** An Luu Phuoc, Tran Minh Duc, Louise Thwaites, Duncan Wagstaff.

**Funding acquisition:** Abi Beane, Rashan Haniffa, Lam Minh Yen, Louise Thwaites, Duncan Wagstaff.

**Investigation:** An Luu Phuoc, Abi Beane, Duncan Wagstaff.

**Methodology:** An Luu Phuoc, Abi Beane, Rashan Haniffa, Lam Minh Yen, Louise Thwaites, Duncan Wagstaff.

**Project administration:** An Luu Phuoc, Abi Beane, Louise Thwaites, Duncan Wagstaff.

**Resources:** An Luu Phuoc, Abi Beane, Duncan Wagstaff.

**Software:** Duncan Wagstaff.

**Supervision:** Vu Dinh Phu, Abi Beane, Rashan Haniffa, Lam Minh Yen, Louise Thwaites, Duncan Wagstaff.

**Validation:** Truong Ngoc Trung, Nguyen Thien Binh, Ha Thi Hai Duong, Thuy Thi Diem Tran, Duong Bich Thuy, Nguyen Thanh Nguyen, Le Thanh Chien, Nguyen Hoang Viet, Bui Ngoc Thanh, Vo Thi Hoang Dung Em, Jennifer Van Nuil, Chau Minh Duc, Huynh Ngoc Hon, Nguyen Le Nhu Tung, Lam Minh Yen, Louise Thwaites, Duncan Wagstaff.

**Writing – original draft:** An Luu Phuoc, Louise Thwaites, Duncan Wagstaff.

**Writing – review & editing:** An Luu Phuoc, Vo Tan Hoang, Truong Ngoc Trung, Nguyen Thien Binh, Vu Dinh Phu, Tran Minh Duc, Ha Thi Hai Duong, Thuy Thi Diem Tran, Duong Bich Thuy, Nguyen Thanh Nguyen, Le Thanh Chien, Doan Bui Xuan Thy, Nguyen Hoang Viet, Bui Ngoc Thanh, Vo Thi Hoang Dung Em, Jennifer Van Nuil, Abi Beane, Rashan Haniffa, Pham Ngoc Thach, Chau Minh Duc, Huynh Ngoc Hon, Nguyen Le Nhu Tung, Lam Minh Yen, Louise Thwaites, Duncan Wagstaff.

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
