## [Decision Letter · Decision Letter 0]

22 Apr 2025

Dear Dr. Wagstaff,

Thank you for submitting your manuscript to PLOS ONE. After careful consideration, we feel that it has merit but does not fully meet PLOS ONE’s publication criteria as it currently stands. Therefore, we invite you to submit a revised version of the manuscript that addresses the points raised during the review process.

We look forward to receiving your revised manuscript.

Kind regards,

Chinh Quoc Luong, MD., PhD.

Academic Editor

PLOS ONE

https://journals.plos.org/plosone/s/file?id=wjVg/PLOSOne_formatting_sample_main_body.pdf   and  and

“This research was funded by Wellcome [(224048/Z/21/Z), (107367/Z/15/Z), (089276/B/09/7),  (217650/Z/19/Z)].”

4. In the online submission form, you indicated that [All data are available upon request.].

6. Please amend the manuscript submission data (via Edit Submission) to include authors Vu Dinh Phu, Doan Bui Xuan Thy, and Pham Ngoc Thach.

7. We notice that your supplementary files are included in the manuscript file. Please remove them and upload them with the file type 'Supporting Information'. Please ensure that each Supporting Information file has a legend listed in the manuscript after the references list.

Additional Editor Comments:

The study evaluated ventilator liberation practices in three Vietnamese ICUs using robust methodologies despite resource constraints. While the work highlights significant limitations impacting care, it provides a valuable foundation for improving medical practices in the region. However, Reviewer 3 noted that the paper would benefit from referencing clinical practice guidelines and discussing how these guidelines could be adapted to Vietnam's unique context, offering a framework for future enhancements in care quality. Please address the comments below and resubmit your manuscript for further consideration.

Reviewers' comments:

Reviewer's Responses to Questions

**Comments to the Author**

1. Is the manuscript technically sound, and do the data support the conclusions?

Reviewer #1: Yes

Reviewer #2: Yes

Reviewer #3: Yes

2. Has the statistical analysis been performed appropriately and rigorously?

Reviewer #1: N/A

Reviewer #2: N/A

Reviewer #3: Yes

3. Have the authors made all data underlying the findings in their manuscript fully available?

Reviewer #1: No

Reviewer #2: Yes

Reviewer #3: Yes

4. Is the manuscript presented in an intelligible fashion and written in standard English?

Reviewer #1: Yes

Reviewer #2: Yes

Reviewer #3: Yes

Reviewer #1: This is a nicely written and thorough analysis of an important clinical issue with important clinical ramifications.

The authors would be commended for the hard work at producing this work. Simultaneously, while the data is useful and has relevant findings for implementation scientists around the world, it is often over-long in the Discussion, which often rehashes many of the points raised in the Results. This could be edited down to enhance readability.

Limitations could also be expanded, so as to focus on some of the sampling issues from the three ICUs, ie very limited nursing staff involved.

Reviewer #2: This study is a very important contribution in developing evidence regarding quality improvement in critical care processes in the LMIC setting. Well done to the authors for this piece of work. The rapid mixed method process evaluation brings out the barriers that are faced in the ICUs to implement the sedation assessment, SAT and SBTs. The contextually sound recommendations to enable the implementation of a QI initiative to improve these processes are practical and potentially feasible.

Reviewer #3: March 27, 2025

To The Editors:

Thank you very much for allowing me to review the article by Luu and coworkers entitled “A rapid evaluation of quality of sedation and ventilation care processes for critically ill patients in Vietnam”. I found this paper to be interesting and informative. The authors used a variety of methods to assess practices in three intensive care units in Vietnam regarding ventilator liberation. I found the methods used to assess practices to be excellent. Resource limitations have a significant impact on ventilator liberation practices in Vietnam. This work not only highlighted the fact that limitations exist, but provided detailed assessments that will allow physicians in this region to improve medical care.

If I were to make a critical comment about this work, it would be that there should be references to clinical practice standards and guidelines in the domain of ventilator liberation. Clearly, such guidelines have been developed only for research-rich environments and could not be applied to intensive care units in Vietnam without modification. However, tools are available to modify existing clinical practice guidelines to regions and nations where there are differences in population demographics and economic situations from those regions for which the original guidelines were developed. This additional commentary could provide a template for future improvements in the quality of care in Vietnam.

**Do you want your identity to be public for this peer review?** For information about this choice, including consent withdrawal, please see our For information about this choice, including consent withdrawal, please see our Privacy Policy .

Reviewer #1: No

Reviewer #2: No

Reviewer #3: **Yes:** Daniel R. Ouellette MDDaniel R. Ouellette MD

While revising your submission, please upload your figure files to the Preflight Analysis and Conversion Engine (PACE) digital diagnostic tool, https://pacev2.apexcovantage.com/ . PACE helps ensure that figures meet PLOS requirements. To use PACE, you must first register as a user. Registration is free. Then, login and navigate to the UPLOAD tab, where you will find detailed instructions on how to use the tool. If you encounter any issues or have any questions when using PACE, please email PLOS at . PACE helps ensure that figures meet PLOS requirements. To use PACE, you must first register as a user. Registration is free. Then, login and navigate to the UPLOAD tab, where you will find detailed instructions on how to use the tool. If you encounter any issues or have any questions when using PACE, please email PLOS at figures@plos.org . Please note that Supporting Information files do not need this step.

---

## [Author Response · Author response to Decision Letter 1]

8 Nov 2025

Many thanks for the opportunity to revise and improve our manuscript.

With regards to the specific reviewer comments:

1. We have amended the manuscript to meets PLOS ONE's style requirements as suggested

2. We have included a complete copy of PLOS’ questionnaire on inclusivity in global research.

3. We have amended the wording to the funding declaration as suggested

4. We have amended the declaration to clarify that all data are available through managed access via the OUCRU Data Access Committee (and provided the website for how to do this). This follows the standard policy of OUCRU, Oxford University and Welcome, to avoid exploitation of LMIC researchers and institutions and enable data compliance with both UK and Vietnamese law.

5. We have amended the manuscript submission data (via Edit Submission) to include authors Vu Dinh Phu, Doan Bui Xuan Thy, and Pham Ngoc Thach.

6. We have uploaded supplementary files as a separate docuemet and included a legend in the manuscript after the references

7. Our reference list is complete and correct.

With regards to additional Editor Comments:

As suggested, we have added a sentence in the Discussion pointing out tools to adapt clinical practice guidelines between settings. We feel this completes the existing extensive discussion of the specific aspects of Vietnamese context which necessitate adaptation of international guidelines.

As suggested, we have added a sentence to stress the limitations on generalisability due to sampling issues

Kind regards,

The authors.

---

## [Editor Report · Decision Letter 1]

2 Dec 2025

A rapid evaluation of quality of sedation and ventilation care processes for critically ill patients in Vietnam

PONE-D-24-29941R1

Dear Dr. Wagstaff,

We’re pleased to inform you that your manuscript has been judged scientifically suitable for publication and will be formally accepted for publication once it meets all outstanding technical requirements.

Kind regards,

Chinh Quoc Luong, MD., PhD.

Academic Editor

PLOS ONE

Additional Editor Comments (optional):

The revised manuscript now fully satisfies the publication standards of PLOS ONE. The authors have comprehensively addressed all editorial and reviewer feedback, resulting in a study that is methodologically rigorous, clearly articulated, and fully compliant with journal policies. Their investigation into sedation and ventilation practices in Vietnamese Intensive Care Units offers valuable evidence for critical care in Low- and Middle-Income Country contexts, and the revisions have further strengthened the study’s clarity, generalizability, and practical significance.
---

## [Editor Report · Acceptance letter]

PONE-D-24-29941R1

PLOS One

Dear Dr. Wagstaff,

I'm pleased to inform you that your manuscript has been deemed suitable for publication in PLOS One. Congratulations! Your manuscript is now being handed over to our production team.

Kind regards,

on behalf of

Assoc. Prof. Chinh Quoc Luong

Academic Editor

PLOS One